# From histology to macroscale function in the human amygdala

Hans Auer[1], Donna Gift Cabalo[1], Raúl Rodríguez-Cruces[1], Oualid Benkarim[1], Casey Paquola[2], Jordan DeKraker[1], Yezhou Wang[1], Sofie Louise Valk[2,3,4], Boris C Bernhardt[1†‡], Jessica Royer[1*†‡]

[1]Montreal Neurological Institute and Hospital, McGill University, Montreal, Canada; [2]Institute for Neuroscience and Medicine, Forschungszentrum Jülich, Jülich, Germany; [3]Max Planck Institute for Human Cognitive and Brain Sciences, Leipzig, Germany; [4]Institute of Systems Neuroscience, Heinrich Heine University Düsseldorf, Düsseldorf, Germany

## eLife Assessment

This **valuable** contribution combines high-resolution histology with magnetic resonance imaging in a novel way to study the organisation of the human amygdala. The main findings **convincingly** show the axes of microstructural organisation within the amygdala and how they map onto the functional organisation. Overall, the approach taken in this paper showcases the utility of combining multiple modalities at different spatial scales to help understand brain organisation.

*For correspondence:
jessica.royer@mail.mcgill.ca

†These authors contributed equally to this work

‡Co-senior authors

Competing interest: The authors declare that no competing interests exist.

**Abstract** The amygdala is a subcortical region in the mesiotemporal lobe that plays a key role in emotional and sensory functions. Conventional neuroimaging experiments treat this structure as a single, uniform entity, but there is ample histological evidence for subregional heterogeneity in microstructure and function. The current study characterized subregional structure-function coupling in the human amygdala, integrating *post-mortem* histology and in vivo MRI at ultra-high fields. Core to our work was a novel neuroinformatics approach that leveraged multiscale texture analysis as well as non-linear dimensionality reduction techniques to identify salient dimensions of microstructural variation in a 3D *post-mortem* histological reconstruction of the human amygdala. We observed two axes of subregional variation in this region, describing inferior-superior as well as mediolateral trends in microstructural differentiation that in part recapitulated established atlases of amygdala subnuclei. Translating our approach to in vivo MRI data acquired at 7 Tesla, we could demonstrate the generalizability of these spatial trends across 10 healthy adults. We then cross-referenced microstructural axes with functional blood-oxygen-level dependent (BOLD) signal analysis obtained during task-free conditions, and revealed a close association of structural axes with macroscale functional network embedding, notably the temporo-limbic, default mode, and sensory-motor networks. Our novel multiscale approach consolidates descriptions of amygdala anatomy and function obtained from histological and in vivo imaging techniques.

## Introduction

The amygdala is a central hub for socio-affective and cognitive functioning (*LeDoux, 2003*; *Adolphs, 1999*; *Pessoa and Adolphs, 2010*). Over the past decades, lesion studies in animals and humans have been crucial in our understanding of this structure's functional role. Studies performed in animal models have reported significant deficits in a vast array of social and affective functions following amygdala lesions, including affective blunting, perturbed social interest and affiliation behaviors,

increased aggression, altered sexual and maternal behaviors as well as fear response to environmental stimuli (*Gothard, 2020*; *Dal Monte et al., 2015*; *Kazama et al., 2012*). In addition to altering social behavior, lesions of this structure in humans have been associated with impaired decision-making (*Hampton et al., 2007*) and deficits in attention and arousal mechanisms (*Sarter and Bruno, 2000*), emphasizing the importance of the amygdala in a broad array of functional domains.

Although earlier work on amygdala function in humans has considered this region as a single, unitary structure, the distinct roles of its individual subdivisions are now increasingly highlighted (*Ball et al., 2007*; *Bzdok et al., 2013*; *Gamer et al., 2010*). Notably, early research on the amygdala in non-human primates has been instrumental in understanding its intricate structure, function, and patterns of anatomical connectivity (*Amaral and Price, 1984*; *Ghashghaei et al., 2007*). This foundational work has highlighted the amygdala's different subdivisions, most notably the basomedial nucleus (BM), basolateral nucleus (BL), and central nucleus (Ce) (*Amaral et al., 1992*). Furthermore, this work describes a dense network between these subdivisions and the prefrontal cortex, most strongly found in the posterior orbitofrontal and anterior cingulate areas.

In humans, qualitative examinations of *post-mortem* specimens have identified several subdivisions within the amygdala, each with distinct cytoarchitectural characteristics and distinguishable connectivity profiles (*Kedo et al., 2018*). These individual subnuclei have often been grouped into larger subdivisions, specifically centromedian, laterobasal, and superficial regions (*Amunts et al., 2005*). Distinguishable connectivity profiles in these subdivisions have been previously observed through the analysis of resting-state functional magnetic resonance imaging (rsfMRI) (*Ghashghaei et al., 2007*; *Caparelli et al., 2017*). This non-invasive technique has been instrumental in interrogating gray matter (GM) connectivity and mapping functional networks in the brain by detecting coordinated hemodynamic signal fluctuations across regions (*Greicius et al., 2009*; *Yeo et al., 2011*; *Biswal et al., 1995*; *Smith et al., 2009*; *Biswal et al., 2010*; *Raichle, 2011*). For instance, previous work has shown synchronized functional signals between the centromedial (CM) subdivision of the amygdala and middle and anterior cingulate cortices, frontal cortex, striatum, insula, cerebellum, and precuneus, supporting processes such as attention control and visceral responses (*Pessoa, 2010*; *Barbour et al., 2010*; *Kapp et al., 1994*). Conversely, the laterobasal (LB) region shows unique connectivity with the inferior and middle temporal gyri and middle occipital gyrus which have been associated with associative processing of environmental information and the integration with self-relevant cognition for decision making (*Pessoa, 2010*; *Winstanley et al., 2004*; *Ghods-Sharifi et al., 2009*; *Boyer, 2008*). The superficial (SF) subdivision of the amygdala has rather been associated with social information processing and social interaction via its unique connectivity to the paracentral lobule, posterior cingulate cortex, and orbitofrontal cortex (*Caparelli et al., 2017*; *Goossens et al., 2009*; *Hurlemann et al., 2009*; *Carr et al., 2003*; *Wicker et al., 2003*). Given these differences across amygdala subregions, combining structural and functional analyses can shed light on the multi-faceted contribution of the amygdala to affective and cognitive functioning by potentially revealing variable participation of its subdivisions in different functional networks.

Our current understanding of the amygdala highlights its multidimensional roles, supported by its complex anatomy and participation in multiple brain networks. However, microstructural atlases of this area developed using quantitative techniques are still lacking but are essential for large-scale investigations of structure-function coupling within this region. Emerging strategies for quantitative segmentations of amygdala subdivisions have shown promising results. For example, a dual-branch convolutional network model trained with features extracted from T1-weighted images could find a strong overlap between automatically segmented labels and a manual segmentation of lateral, basal, cortico-superficial and centromedial subregions (*Liu et al., 2020*). Similarly, a machine learning-based correction model could achieve comparable accuracy using a multi-atlas segmentation model (*Hanson et al., 2012*). Despite the promise of these methods, they remain to be validated in histology (*Amunts and Zilles, 2015*), which remains a key technique to validate MRI-based features with access to ground-truth measures of cytoarchitecture (*Yang et al., 2013*; *Alkemade et al., 2023*; *Amunts et al., 2020*). Indeed, regions with different cytoarchitectures often show distinct myelination patterns, which can be observed through various MRI contrasts. Such myelin-sensitive imaging contrasts can differentiate regions with distinct intracortical myeloarchitectonic profiles, demonstrating ties between variations in cellular architecture and myelin distribution (*Ganzetti et al., 2014*; *Baxi et al., 2022*).

Our study seeks to elucidate the intricate structure-function relationships within the amygdala by leveraging advanced data-driven quantitative methods and high-resolution histology. Our approach first leveraged computer vision techniques to map major subdivisions of the amygdala in BigBrain (*Amunts et al., 2013*), a *post-mortem*, high-resolution 3D histological dataset providing direct measurements of brain cytoarchitecture. We translated this approach to in vivo MRI data acquired at ultra-high fields enabling individual-specific assessments of microstructure-function coupling in the amygdala. Harnessing myelin-sensitive contrasts and multi-echo rsfMRI, our study identifies a principal axis of microstructural and functional network dissociation within the human amygdala from a data-driven analysis of its cyto- and myeloarchitecture.

## Results

### Data-driven histological analysis of the human amygdala

Using a radiomics approach (*Schleicher et al., 1999*; *Palomero-Gallagher and Zilles, 2018*), we computed the four central moments (mean, variance, skewness, and kurtosis) from cell body staining intensities in the amygdala provided by the BigBrain dataset (*Amunts et al., 2013*; *Xiao et al., 2019*; *Paquola et al., 2021*; *Figure 1A*). These voxel-wise metrics were computed across different kernel sizes, with local three-dimensional neighborhoods ranging from a radius of 2–10 voxels. As expected, increasing kernel values produced smoother images for all central moments. Thus fine-grain features were emphasized at small kernel values and coarser features with larger kernel values.

The resulting feature bank showed heterogeneous feature profiles across selected moments and kernel sizes (*Figure 1B*). Mean intensities smoothly increased in the ventral to dorsal direction, culminating in the highest intensity values in the dorsal subnucleus regions, and mainly coinciding with the CM subdivision of the amygdala. Variance, however, was highest along the amygdala's borders with the entorhinal cortex and in the amygdala-striatum transition zone. Skewness and kurtosis maps generally co-varied spatially and both highlighted high skewness and kurtosis within the lateral nucleus. Together, these findings indicate that the selected features captured both unique and shared characteristics of histological signal variations within the amygdala.

We then applied Uniform Manifold Approximation and Projection (UMAP), a non-linear dimensionality reduction technique that preserves the local and global structure of high-dimensional data (*Figure 1C*) by projecting it into a lower-dimensional space (*Becht et al., 2018*) to derive a two-dimensional embedding of amygdala cytoarchitecture (*Figure 1D*). This approach allowed us to bring the high-dimensional histological feature space to a 2D embedding space composed of every amygdala voxel. As such, amygdala voxels were ordered along two dimensions, U1 and U2, capturing two axes of variance in amygdala cytoarchitecture. To better visualize which features may be driving each UMAP dimension, we sorted all input features along U1 and U2 (*Figure 1E*). Ordering the mean along U1 highlighted increasing intensity values at all different kernel sizes, where the highest intensity voxels co-localized with low values in U1. While the variance, skewness, and kurtosis showed less systematic changes at lower kernel values, patterns became more evident at higher kernels. Indeed, variance and kurtosis seemed to show an opposite trend to the mean along U1, where higher skewness and kurtosis co-localized more strongly with positive values of U1 (*Supplementary file 1a*). In contrast, sorting the feature matrix by U2 showed very similar trends between all features, independent of its moment and kernel value. More specifically, the highest intensity voxels were mostly found to show higher values along U2 (*Supplementary file 1b*). Overall, these UMAP-driven visualizations of the histological feature space suggest our dimensionality reduction approach could recover moment-specific (U1) as well as global intensity covariations across moments (U2).

To contextualize variations in U1 and U2 values across the amygdala, we computed voxel-wise correlations between each UMAP component and the amygdala coordinate space. U1 primarily varied along the inferior-superior axis (U1: $r=0.8340$; U2: $r=-0.0137$), followed by posterior-anterior (U1: $r=-0.5282$; U2: $r=0.2793$) and medial-lateral directions (U1: $r=0.1399$; U2: $r=-0.2657$) (*Figure 1G*). Statistical significance of correlations was assessed using a variogram matching approach (*Burt et al., 2020*) implemented in the BrainSpace toolbox (*Vos de Wael et al., 2020*; *Figure 1G*). Both UMAP components were found to significantly co-vary along the posterior-anterior axis (U1: $p_{null} < 0.001$; U2: $p_{null} = 0.002$), while only U1 was significantly correlated with the inferior-superior axis (U1: $p_{null} < 0.001$;

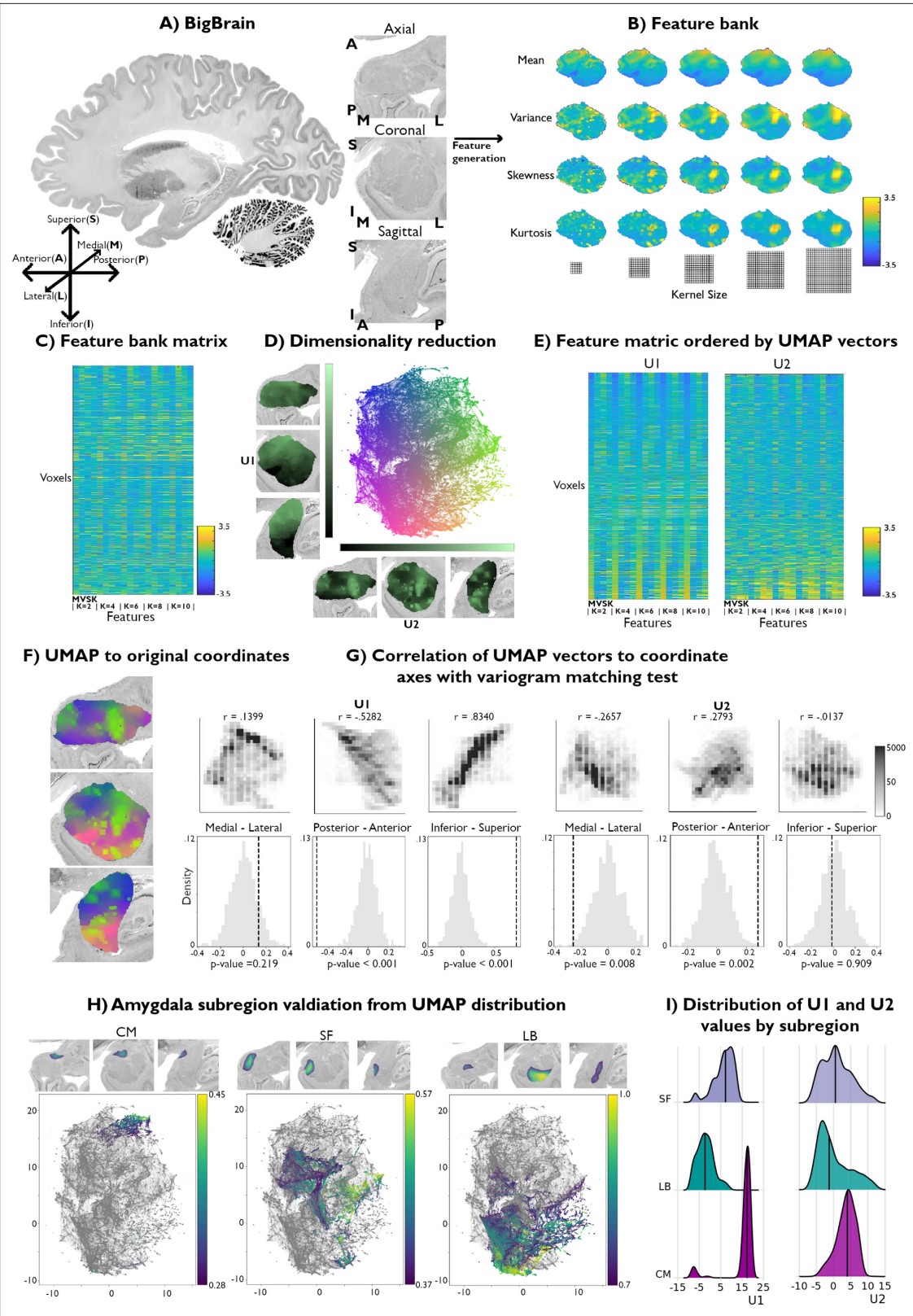

**Figure 1.** Data-driven histological mapping of the human amygdala. (**A**) The amygdala was segmented from the 100 micron resolution BigBrain dataset using an existing subcortical parcellation (**Xiao et al., 2019**). Slice orientation of subpanels containing amygdala images is consistent across all panels in this figure. (**B**) Leveraging the pyRadiomics package v3.0.1 (**van Griethuysen et al., 2017**), we built a multiscale histological feature bank of the amygdala capturing fine-to-coarse intensity variations within this structure. Feature values were all normalized to better visualize relative intensity

*Figure 1 continued on next page*

*Figure 1 continued*

differences. (**C**) Matrix representation of the normalized feature bank shown in *A*. (**D**) We applied Uniform Manifold Approximation and Projection (UMAP) to this feature bank to derive a low-dimensional embedding of amygdala cytoarchitecture, defining a two-dimensional coordinate space (scatter plot, middle). Colors of the scatter plot represent proximity to axis limits. (**E**) Reordering the feature bank according to each eigenvector (**U1 and U2**) highlights the underlying variance in each feature captured by UMAP. (**F**) Coloring each amygdala voxel according to its corresponding location in the UMAP embedding space partially recovered its anatomical organization. (**G**) U1 and U2 were correlated to the three spatial axes and variogram matching tests assessed the statistical significance of each correlation (statistical significance threshold set at p<0.05). (**H**) Coloring the embedding space with openly available probabilistic map labels of the three main amygdala subregions showed that this region's microstructural architecture could be recovered by UMAP. (**I**) Ridge plots of the probability values per subregion also illustrate a characterization of the subregions in U1.

The online version of this article includes the following figure supplement(s) for figure 1:

**Figure supplement 1.** Replication of data-driven histological mapping for the right amygdala.

---

U2: $p_{null}$ = 0.909). Neither U1 nor U2 were significantly correlated with the medial-lateral coordinate (U1: $p_{null}$ = 0.441; U2: $p_{null}$ = 0.068).

## Validation of histological space using independent post-mortem dataset

We contextualize our data-driven approach using established probability maps of amygdala microstructure. These openly available probability maps generated from visual inspection of 10 *post-mortem* brains divide the amygdala into CM, SF, and LB subregions (*Amunts et al., 2020*). Maps were thresholded to retain only voxels with the highest 5% probability values and binarized. Plotting the probability values of retained voxels for each subregion showed that UMAP could dissociate these established cytoarchitectural subdivisions of the amygdala (*Figure 1H*). The three subregions could be particularly segregated along the U1 component. Indeed, we found significant differences in U1 values across the three amygdala subdivisions (*F*=1630.8; $p_{null}$ <0.001), while no significant difference was found across U2 (*F*=30.3; $p_{null}$ <0.581) (*Figure 1I*). Together, these findings show our theoretically grounded framework can successfully distinguish different subregions in the amygdala in a purely data-driven way. Additionally, results could be replicated when analyzing signals from the right amygdala, supporting the potential generalizability of our framework (*Figure 1—figure supplement 1*).

## In vivo generalizability of histological space

We also assess the generalizability of these results to in vivo myelin-sensitive MRI data. We leverage quantitative T1 imaging collected at a field strength of 7 Tesla (7T) in 10 unrelated, healthy participants (*Figure 2A*). These images offered a resolution of 500 µm and were run through a similar analytical framework as the BigBrain dataset. We used individual-specific segmentations of the amygdala obtained with VolBrain (*Manjón and Coupé, 2016*). Once again, a feature bank was rendered from the same central moments, specifically mean, variance, skewness, and kurtosis. Kernel sizes varied from size 1–5, resulting in 20 distinct feature maps (*Figure 2A*). The new feature bank was again submitted to UMAP for dimensionality reduction (*Figure 2A*), and the values of each component were plotted back to their respective coordinates in the amygdala.

We then compared spatial variations of the two UMAP components uncovered in each participant with the UMAP space derived from the BigBrain dataset. When examining the spatial layout of the UMAP components, we find similar neuroanatomical trends in all subjects to those found in histology (*Figure 2B*). Notably, the U1 vector of all subjects are found to be significantly correlated to the inferior-superior axis from a variogram matching test (*Supplementary file 1c*), similar to BigBrain, suggesting the potential for this framework to capture important structural features in histology as well as in vivo MRI. We also find that the medial-lateral axis correlations to U1 across all subjects are consistent with the variogram test (*Supplementary file 1c*). The spatial layout of U2, on the other hand, showed lower consistency across participants, as none of the coordinate axes were significantly correlated with U2 in more than 7/10 subjects (*Supplementary file 1d*). In sum, we identify a single axis (U1) able to pick up on important amygdala microstructural features in both *post-mortem* histology and in vivo markers of GM microstructure.

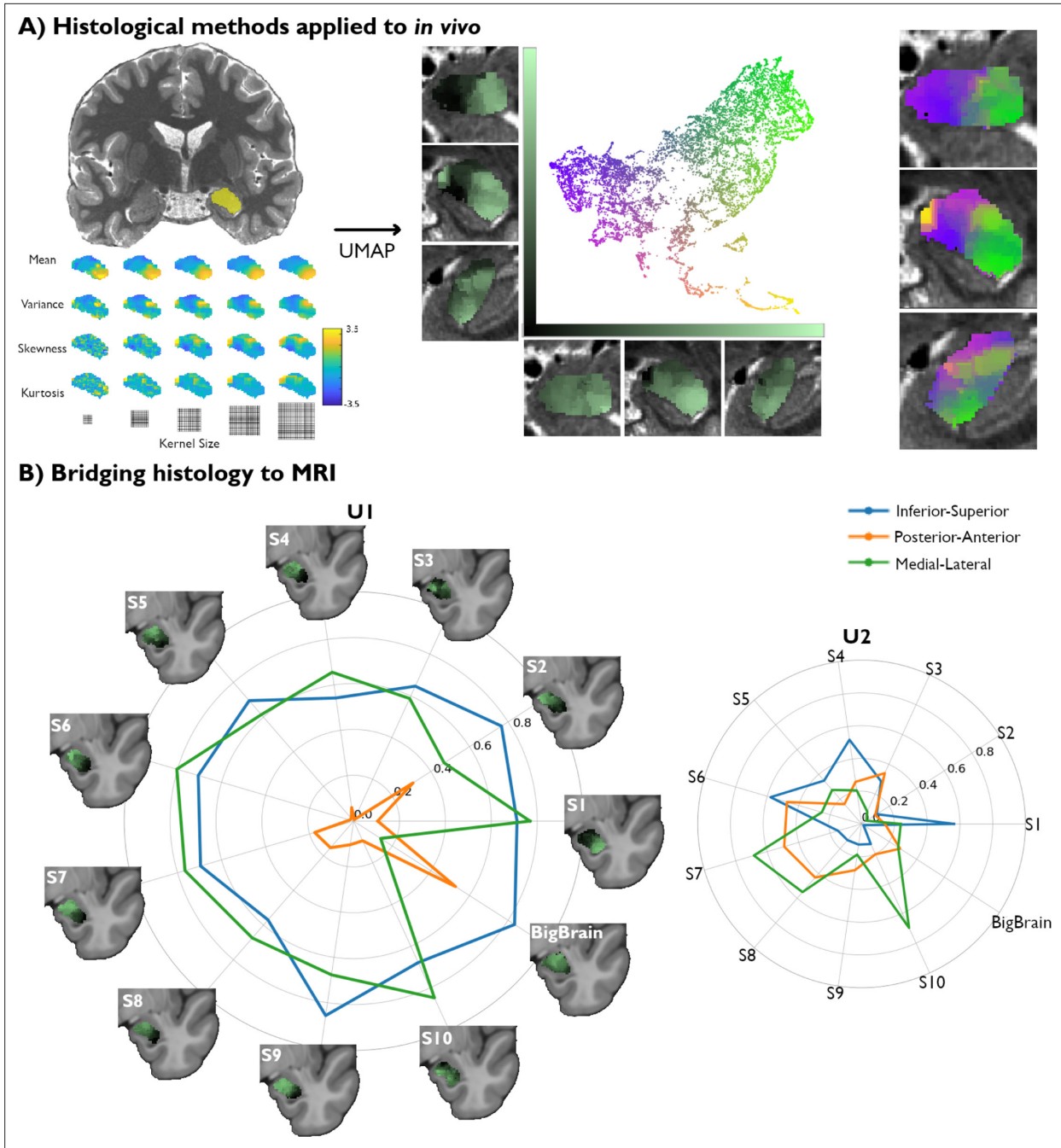

**Figure 2.** Translating amygdala histological space to in vivo, ultra-high-resolution, myelin-sensitive magnetic resonance imaging (MRI). (**A**) We segmented the left and right amygdalae of individual subjects from quantitative T1 (qT1) scans, and applied the same framework as developed in *post-mortem* imaging to derive subject-specific, in vivo representations of amygdala microstructure. (**B**) Correlation values between the Uniform Manifold Approximation and Projection (UMAP) components (U1 and U2) and the three coordinate axes of the 10 MRI subjects were computed in MNI152 space and then contrasted with the correlation values found in the histological data (BigBrain transformed to MNI152 space).

## Association with macroscale function

We next sought to investigate associations between amygdala microstructural organization and this region's macroscale functional organization across subjects. Given U1's strong spatial consistency across participants and microstructural modalities, we generated subject-specific masks segregating this component's highest and lowest 25% of values within the amygdala for each participant. This resulted in two distinct regions of interest, reflecting the anchors of maximal microstructural

dissociation within the amygdala for each participant, from which we could extract corresponding functional activity recorded at rest (*Figure 3A*).

Voxel-wise rsfMRI time series were averaged within each microstructural subregion and correlated to vertex-wise cortical time series. A linear mixed effect model comparing amygdalo-cortical connectivity profiles between both subregions showed that functional network affiliations significantly differed across U1 subregions, with stronger connectivity observed between the superior portions of the amygdala (top 25% U1 values) and the prefrontal lobe (*Figure 3B*). Stratifying functional connectivity patterns of each amygdala subregion to the cortex according to established intrinsic functional network communities further highlighted the relatively stronger connectivity of the superior subregions to all cortical networks, particularly the limbic, frontoparietal, and default mode networks (*Figure 3C*). Meta-analytical decoding of subregional connectivity profiles using NeuroSynth (*Yarkoni et al., 2011*) emphasized the functional dissociation in connectivity patterns of both microstructurally-defined areas. This analysis showed that the functional connectivity pattern of the region with the highest 25% U1 values was most strongly associated with terms relating to autobiographical memory ('autobiographical' and 'autobiographical memory'), while the other seed region's connectivity profile overlapped with activation patterns related to emotional input ('happy faces,' 'neutral faces,' and 'fearful faces') (*Figure 3C*). Furthermore, decoding our statistical effects map (Region 1 connectivity >Region 2 connectivity) highlighted associations with terms relating to the self, introspection, and reward ('self-referential,' 'referential,' 'moral,' 'autobiographical,' 'smoking,' 'craving'). Collectively, these findings show that our theoretically grounded approach, developed in histology and generalizable to microstructurally-sensitive in vivo MRI data, can delineate distinct functional network embeddings in the human amygdala.

## Discussion

The amygdala is a crucial structure for several aspects of cognitive and socio-affective functioning (*Pessoa and Adolphs, 2010*). These functions are supported by complex connectivity patterns to other brain regions, stemming from distinct subnuclei with unique microstructural properties (*Kedo et al., 2018*). However, current investigations of structure-function coupling in the amygdala are limited by a lack of datasets and tools for individualized and observer-independent delineation of its subregions. Indeed, the strong inter-individual variability of its structural and functional organization (*Amunts et al., 2005*; *Sylvester et al., 2020*) motivates more personalized approaches to reliably study the microstructural determinants of amygdala function and connectivity. In the current paper, we present a data-driven exploration of subcortical cytoarchitecture applied to the human amygdala. We could translate this approach to microstructurally-sensitive in vivo MRI data as a bridge, to ultimately examine associations between microstructural subregions and functional networks. As such, the present work defines a quantitative and integrated account of the amygdala's microstructural composition and functional organization. In doing so, our approach sets the stage for novel investigations spanning other subcortico-cortical systems, and shows potential to deliver new insights into brain-wide principles of structure-function coupling and how this interplay may be altered in clinical populations.

The proposed framework aimed to delineate amygdala subnuclear organization by leveraging a multiscale texture processing pipeline designed to retain finer and coarser regional cytoarchitectonic properties. We specifically harness radiomics, a field with established diagnostic and prognostic potential in medical imaging (*Limkin et al., 2017*). Feature selection in our study was motivated by previous work conducted at the level of the neocortex (*Amunts and Zilles, 2015*; *Schleicher et al., 1999*) and focused on the four central moments, specifically, the mean, variance, skewness, and kurtosis of voxel subsets, to reflect regional texture variability related to amygdala cytoarchitecture. In contrast to qualitative approaches based on single features, such as investigations based on the detection of specific cell types (*Bonin, 1961*), our pipeline captures several aspects of amygdala microstructure informing non-linear dimensionality reduction methods applied to a high-dimensional feature space. The resulting components U1 and U2 reflected complex combinations of central moments, with U1 being mainly scaled to mean intensities and U2 being associated with weighted combinations of the different moments. Crucially, we validated this coordinate space using openly available maps of amygdalar subdivisions from histological examinations performed by expert neuroanatomists (*Amunts et al., 2020*). This approach complements previous work harnessing subnuclear parcellations

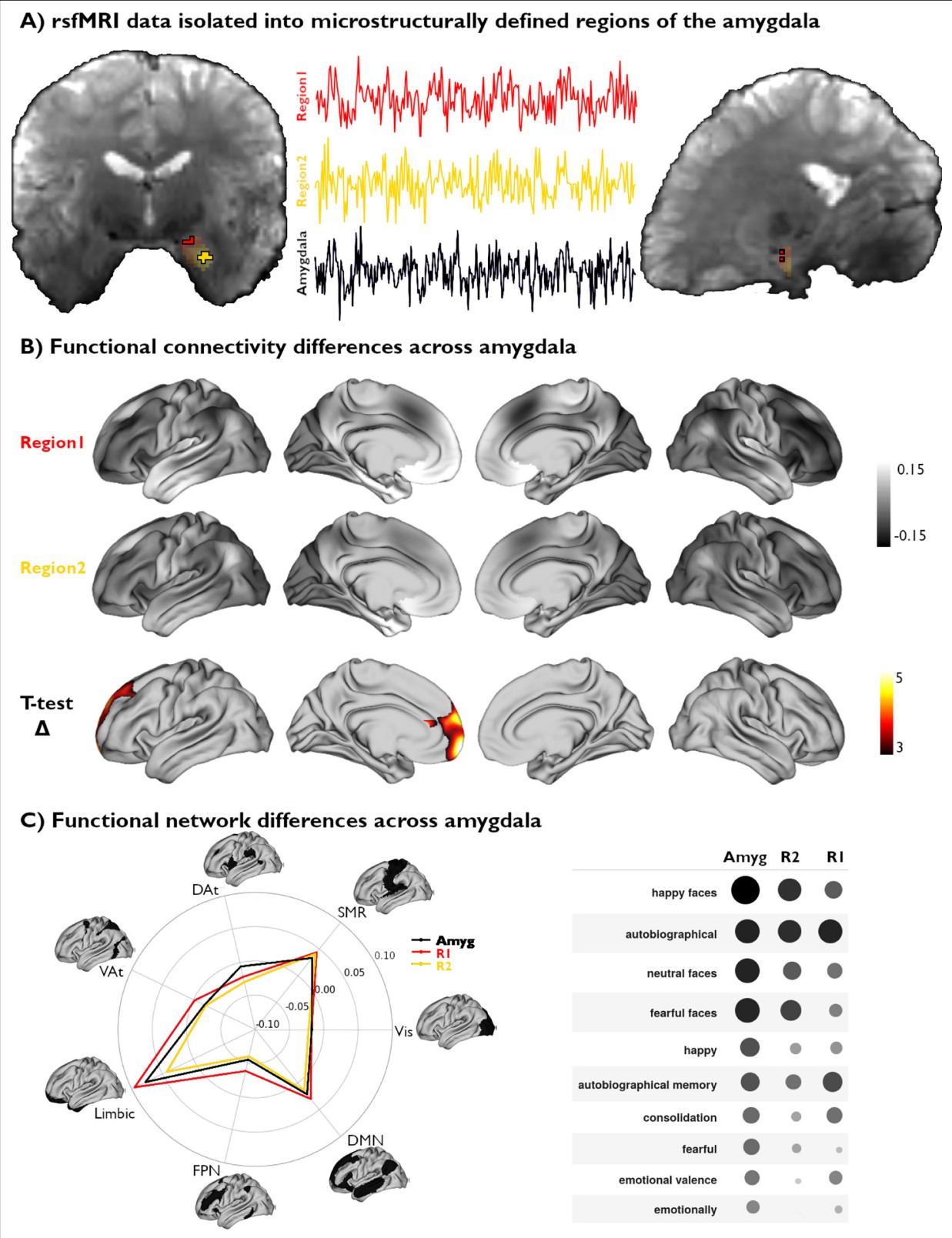

**Figure 3.** Functional network mapping of amygdala microstructural subregions. (**A**) We isolated the resting-state functional magnetic resonance imaging (rsfMRI) time series of two amygdala subregions, defined from subject-specific U1 topography, as well as the whole amygdala. (**B**) We computed the functional connectivity of both amygdala subregions, and project resulting correlations to the cortex. We further demonstrate the differences in connectivity patterns between both subregions (t-value) and highlight the regions with significant differences (*pFWE*<0.05). (**C**) Left:

*Figure 3 continued on next page*

*Figure 3 continued*

The activation patterns illustrated in (B, top) were averaged within intrinsic functional communities defined by *Yeo et al., 2011*. Right: Meta-analytic decoding of functional connectivity patterns of both amygdala subregions and the whole amygdala dissociated cognitive and affective functional affiliations of this region.

The online version of this article includes the following figure supplement(s) for figure 3:

**Figure supplement 1.** Functional network mapping of amygdala microstructural subregions.

of the amygdala derived from visual inspections of post-mortem specimens (*Kedo et al., 2018*) and deep-learning algorithms applied to in vivo MRI data (*Liu et al., 2020*; *Liu et al., 2019*; *Saygin et al., 2017*) to contextualize variations in functional connectivity profiles within this region. Indeed, a significant advantage of our framework lies in the ease with which it may be applied to new datasets. For instance, our method overcomes the time-consuming nature and high level of expertise required for precise manual subnuclear segmentations. Furthermore, this approach circumvents the need for large datasets required to validate deep learning-based applications, a particular concern in the case of histological data which are often limited to few or even single specimens. However, it is important to note that both datasets analyzed in this work are limited by their small sample size (n=1 for BigBrain and n=10 for MICA-PNI). We speculate that the signal variations captured by U2 may be sensitive to artifacts or subject-specific sources of variance, potentially explaining why it was not consistent between subjects and modalities. Moving forward, this hypothesis could be assessed in future work via the analysis of larger histological and neuroimaging datasets to better track recurring features picked up by U2 or the association of these unique topographies with behavioral markers. Overall, the proposed framework lays the groundwork for future investigations of subcortical structure-function coupling by anchoring connectivity and task-related activations within measurements of regional microarchitecture.

The present works described a continuous coordinate space of amygdala subregional microstructure. However, this region has been previously described as consisting of three main subdivisions: LB, CM, and SF, each composed of smaller subnuclei with distinct connectivity patterns and functions (*Ball et al., 2007*; *Bzdok et al., 2013*; *Amunts et al., 2005*; *de Olmos and Heimer, 1999*). These subregions are largely conserved between humans and monkeys, reflecting their evolutionary relationship. However, there are still some considerable differences such as in the SF subregion, where its description in monkeys additionally contains the lateral olfactory tract (LOT) (*Olmos, 1990*). Although qualitative histological accounts have indeed delineated multiple subunits within these general regions, the present work focuses on three subdivisions (*Amunts et al., 2005*). To account for resolution disparities when translating our findings to in vivo MRI data. The LB subdivision includes the basomedial nucleus (Bm), basolateral nucleus (BL), lateral nucleus (LA), and paralaminar nucleus (PL). Moving medially, the CM subdivision includes the central (Ce) and medial nuclei (Me), while the SF subdivision includes the anterior amygdaloid area (AAA), amygdalohippocampal transition area (AHi), amygdalopiriform transition area (APir), and ventral cortical nucleus (VCo) (*Heimer et al., 1999*). However, disagreement on the precise attribution of nuclei to broader subdivisions motivated our investigations of probabilistic subunits of the amygdala (*Kedo et al., 2018*). The development of new tools to segment amygdala subnuclei in vivo opens opportunities for future work to further validate our framework at the precision of these nuclei within subjects (*Saygin et al., 2017*). We selected UMAP for its potential to recover this nuclear architecture via the identification of discrete clusters of microstructural similarity within the amygdala. While these dimensions partially align with traditional concepts of arealization, they also provide a complementary, graded representation of amygdala microarchitecture. Although our framework leverages ultra-high resolution histological and myelin-sensitive MRI, the inherent spatial autocorrelation of feature intensities in these modalities may have emphasized the continuous signal variations we identify within the amygdala and hindered the discovery of discrete boundaries between known subdivisions. We address this limitation by benchmarking U1 and U2 distributions against validated probabilistic maps of major amygdala subdivisions (*Amunts et al., 2005*; *Amunts et al., 2020*), enabling us to recover the established biological validity of its nuclear organization. Furthermore, this approach allowed us to derive discrete clusters of maximal microstructural differentiation within the amygdala; these clusters served as seed regions for microstructurally-grounded and individualized investigations of the functional connectome embedding of amygdala subregions. Following an inferior-superior and medial-lateral axis of differentiation in both *post-mortem* histology

and myelin-sensitive in vivo MRI, this bipartite division is in line with previous work investigating the structural connectivity of the amygdala using diffusion-weighted imaging and probabilistic tractography (*Bach et al., 2011*) as well as functional connectivity from rsfMRI (*Mishra et al., 2014*). Indeed, both modalities highlight the existence of two distinct clusters segregating amygdala connectivity to temporopolar and orbitofrontal cortices. These findings mirror macroscale associations seen in the neocortex between microstructure and connectivity, which emphasize close correspondence between the strength of interareal connectivity and microstructural similarity (*Barbas, 2015*; *García-Cabezas et al., 2019*; *Barbas, 1986*). In the case of the amygdala, our framework could thus recover these distinct anatomical pathways from a data-driven, texture-based analysis of microarchitectural information alone, supporting the potential of such contrasts to provide insights into the large-scale network embeddings of subcortical systems.

In line with this suggested association between amygdala microarchitecture and functional connectivity, we conclude our analyses by leveraging subject-specific representations of amygdala microstructure to map large-scale variations in its functional connectivity to the neocortex. We isolated and contrasted the highest and lowest 25% of U1 values for each participant to define an individualized bipartite parcellation of the amygdala. Qualitatively, we found that the subregion defined by the highest 25% of U1 values mainly overlapped with what is commonly defined as the superficial and centromedial subregions, whereas the lowest 25% U1 values subregion overlapped mostly with the laterobasal division. Interestingly, CM and SF-characterized subregions showed significantly stronger functional connectivity to prefrontal structures. This finding aligns with previous work demonstrating unique affiliations between the CM subregion and anterior cingulate and frontal cortices (*Barbour et al., 2010*; *Kapp et al., 1994*), as well as between the SF subregion and the orbitofrontal cortex (*Caparelli et al., 2017*; *Pessoa, 2010*; *Goossens et al., 2009*; *Klein-Flügge et al., 2022*). Although these findings are promising, we also observe considerable overlap between functional connectivity networks of both our defined subregions. Indeed, the amygdala is a relatively small structure, leading to likely interconnectivity between its subregions and locally high signal autocorrelation. Functional connectivity and microstructure in the amygdala are certainly related, however, previous work suggests they do not perfectly overlap (*Bzdok et al., 2013*). In addition, this region is affected by a relatively low signal-to-noise ratio (SNR), as is observed in broader temporobasal and mesiotemporal territories. Decoding of subregional functional connectivity results indicated possible dissociations in cognitive (e.g. memory) and affective (e.g. emotional face processing) functions of the amygdala, echoing previous accounts of this region's functional specialization and subregional segregation of associative processing of emotional stimuli. Notably, these findings link the functional connectivity profile of a subregion partially co-localizing with LB to emotional face processing. The LB subregion has been previously linked to associative processing related to the integration of sensory information (*Bzdok et al., 2013*; *Pessoa, 2010*; *Winstanley et al., 2004*; *Ghods-Sharifi et al., 2009*; *Boyer, 2008*), which is consistent with the association with visual-emotional information processing identified in the present work. For the right amygdala, dissociation in functional connectivity patterns were more subtle, leading to overall similar functional decoding across the two clusters (*Figure 3—figure supplement 1*). Overall, our findings suggest that this microstructurally-grounded delineation of U1 subregions could capture dissociations in their respective functional associations and potentially with fear-related processes. These results echo previous chemoarchitectural descriptions of the amygdala involving the 5-HT receptor, which has been closely associated with fear responses in mice and humans (*Hurlemann et al., 2009*; *Ramboz et al., 1998*). Indeed, this receptor is expressed in lower densities in the CM region, overlapping with our U1 subregions that show lower connectivity to regions involved in fear-related responses. The present work thus offers an important step towards a more integrated account of the amygdala's microstructural composition and functional organization.

By harnessing an openly available arsenal of tools and methods from histology, radiomics, and neuroinformatics, we define a comprehensive framework that enhances our understanding of individual differences in amygdala organization. Our findings contribute to a growing body of research emphasizing the importance of integrating precise structural and functional in vivo measures to elucidate the complex roles of the both subcortical and cortical regions, in both health and disease (*Paquola et al., 2022*; *Paquola et al., 2025*; *Royer et al., 2023*; *DeKraker et al., 2024*). This multimodal, multiscale, and subject-specific approach not only advances our knowledge of the amygdala's microstructural and functional intricacies but also offers a valuable resource for future studies

exploring subcortical structures and their implications in various neurological and psychiatric conditions. This integrated perspective is essential for developing more precise and personalized interventions for disorders associated with amygdala dysfunction.

## Methods

### Histological data acquisition and pre-processing

Cell-body-staining intensity of the amygdala was obtained from the BigBrain dataset (*Amunts et al., 2013*). BigBrain is an ultra-high–resolution Merker-stained 3D volumetric histological reconstruction of a *post-mortem* human brain from a 65-year-old male, made available on the open-access BigBrain repository (bigbrain.loris.ca). This data was acquired through the body donor program of the University of Düsseldorf in accordance with legal requirements. The *post-mortem* brain was paraffin-embedded, coronally sliced into 7400 20 µm sections, silver-stained for cell bodies (*Ball et al., 2007*), and digitized. As such, image intensity values in this dataset provide direct measurements of brain cytoarchitecture. Following manual inspection for artifacts, automatic repair procedures were applied, involving nonlinear alignment to a *post-mortem* MRI, intensity normalization, and block averaging (*Paquola et al., 2019*). 3D reconstruction was implemented with a successive coarse-to-fine hierarchical procedure. All main analyses were performed using the 100 µm isovoxel resolution dataset.

### Amygdala segmentation and subdivision mapping

The left and right amygdalae were isolated from the BigBrain volume using an existing manual segmentation of left and right subcortical structures (*Xiao et al., 2019*). This segmentation was warped to BigBrain histological space from the standard ICBM2009b symmetric template space (*Fonov et al., 2011*) using openly available co-registration strategies aggregated in the BigBrainWarp toolbox (*Xiao et al., 2019*; *Paquola et al., 2021*). Notably, these approaches were optimized to improve the alignment of subcortical structures (*Xiao et al., 2019*). After registering the subcortical atlas to histological space and resampling the segmentation to an isovoxel resolution of 100 µm, we generated unique binary masks isolating the left and right amygdalae and performed manual corrections on each mask (i.e. improving smoothness and continuity of the mask borders), and eroded the mask by five voxels to provide a conservative estimate of regional borders. All main analyses were performed on left hemisphere data only, while the right hemisphere served as a validation dataset (*Figure 1—figure supplement 1*).

To contextualize our data-driven histological mapping of the amygdala (see below), we leveraged openly available probabilistic maps of amygdala subnuclei derived from visual inspections of *post-mortem* tissue specimens performed by expert neuroanatomists (*Kedo et al., 2018*; *Amunts et al., 2020*). The borders of amygdala subdivisions were traced in 10 *post-mortem* brains. Following 3D reconstruction and alignment of each *post-mortem* brain to a common template space, voxel-wise probabilistic maps for each subregion were computed by quantifying the consistency of label assignments across the 10 donor brains. For the present work, all available probabilistic maps of the amygdala, including large subdivision groups encompassing multiple amygdala subnuclei (i.e. CM, LB, and SF subdivisions) were accessed from the EBrains repository (v8.2) (*Amunts et al., 2020*). Regional probabilistic maps were warped from ICBM2009c asymmetric space to the ICBM2009b symmetric template using the SyN algorithm implemented in the Advanced Normalization Tools software (ANTs) (*Avants et al., 2008*). Each subregional probabilistic map was subsequently warped to BigBrain histological space (*Xiao et al., 2019*; *Paquola et al., 2021*) and was resampled to an isovoxel resolution of 100 µm. We then generated a maximum probability map to parcellate the amygdala into its subdivisions by retaining the voxels with the highest 5% probability values of belonging to each subdivision.

### Histological feature extraction

We built a histological feature bank of amygdala cell-body-staining intensities using methods from the field of radiomics (*Zhou et al., 2018*). Our approach for feature selection was also inspired by quantitative cytoarchitectural analyses developed in foundational neuroanatomical studies (*Schleicher et al., 1999*), involving the parameterization of intensity profiles with four central moments to characterize regional cytoarchitecture across the neocortex. In the present work, we computed these same four central moments (i.e. mean, variance, skewness, and kurtosis) of cell body staining

intensities in the amygdala to characterize intensity differences across this region. We used pyRadiomics v3.0.1 (*van Griethuysen et al., 2017*) to compute voxel-based maps for each of the selected first-order features, varying the size of 3D-feature extraction to a voxel neighbourhood of 500 μm to 2100 μm (in 400 μm increments). Outlier values (>1 standard deviation from the mean intensity value) were excluded from moment calculations at each kernel size. This resulted in 20 distinct feature maps, capturing variations in intensity distributions within the amygdala at finer and coarser scales. These feature maps were normalized by z-scoring each feature at each kernel size.

## Dimensionality reduction of histological features

To capture and visualize the underlying structure of amygdala cytoarchitecture, we applied UMAP to our normalized histological feature bank (*Becht et al., 2018*). This algorithm was selected over other compression approaches for its scalability in the analysis of large datasets, as well as its ability to preserve both local and global data structures (*Kobak and Linderman, 2021*). Two UMAP hyperparameters controlling the size of the local neighbourhood as well as the local density of data points were kept at their default settings ($n_{neighbours}$ = 15, $dist_{min}$ = 0.1). The resulting low-dimensional embedding of higher-order histological features was contextualized in relation to the amygdala's $x$, $y$, and $z$ voxel coordinate space, and validated against the previously described maximum probability map of the amygdala.

## In vivo MRI data acquisition

After establishing this framework with *post-mortem* histological data, we assessed its generalizability to in vivo, myelin-sensitive MRI contrasts. We capitalized on quantitative T1 (qT1) relaxometry data collected from 10 participants at a field strength of 7 Tesla. This sequence has been shown to be sensitive to cortical myeloarchitecture (*Lutti et al., 2014*; *Stüber et al., 2014*; *Waehnert et al., 2016*), and could thus offer complementary insights into the microstructural organization of the amygdala. Our cohort of 10 adult participants (5 men, mean ± SD age=27.3 ± 5.71 years) were all healthy, with no history of neurological or psychiatric conditions (*Cabalo et al., 2024*). MRI data acquisition protocols were approved by the Research Ethics Board of McGill University. All participants provided written informed consent, which included a provision for openly sharing all data in anonymized form.

Scans were acquired at the McConnell Brain Imaging Centre (BIC) of the Montreal Neurological Institute and Hospital on a 7T Terra Siemens Magnetom scanner equipped with a 32-receive and 8-transmit channel head coil. Synthetic T1-weighted (UNI) and quantitative T1 relaxometry (qT1) data were acquired using a 3D-MP2RAGE sequence (0.5 mm isovoxels, 320 sagittal slices, TR=5170 ms, TE=2.44 ms, TI1=1000 ms, TI2=3200 ms, flip angle$_1$=4°, iPAT=3, partial Fourier=6/8 flip angle$_2$=4°, FOV = 260 × 260 mm$^2$). We combined two inversion images to minimize sensitivity to B1 inhomogeneities and optimize reliability (*Haast et al., 2016*; *Marques et al., 2010*). rsfMRI scans were acquired using a multi-echo, 2D echo-planar imaging sequence (1.9 mm isovoxels, 75 slices oriented to AT-PC-31 degrees, TR=1690 ms, TE1=10.8 ms, TE2=27.3 ms, TE3=43.8 ms, flip angle=67°, multiband factor=3). The rsfMRI scan lasted ~6 min, and participants were instructed to fixate a cross displayed in the center of the screen, to clear their mind, and not fall asleep.

## Multimodal MRI processing and analysis

### Anatomical segmentation and co-registration

Image processing leading to the extraction of cortical and subcortical features and their registration to surface templates was performed via *micapipe v0.2.3*, an open multimodal MRI processing and data fusion pipeline (https://github.com/MICA-MNI/micapipe/ copy archived at *Rodríguez-Cruces and Royer, 2025*; *Cruces et al., 2022*). The amygdala was automatically segmented on the T1w images with volBrain v3, in every subject (*Manjón and Coupé, 2016*). Cortical surface models were generated from MP2RAGE-derived UNI images using FastSurfer 2.0.0 (*Henschel et al., 2020*; *Henschel et al., 2022*). Surface extractions were inspected and corrected for segmentation errors via the placement of manual edits. Native-surface space cortical features were registered to the fs-LR template surface using workbench tools (*Van Essen et al., 2012*).

## qT1 image processing and analysis

Automated, individual-specific segmentations of the amygdala were obtained with VolBrain, and were manually inspected and corrected prior to image processing. In a similar fashion to the histological data, a feature bank was rendered from the same central moments, although kernel sizes varied from size 1–5, rather than 2–10 in order to take into account the difference in image resolution between the two datasets. Thus, voxel neighborhoods ranged from 1500 µm to 5500 µm (in 1000 µm increments). This resulted in 20 distinct feature maps, capturing variations in intensity distributions within the amygdala at finer and coarser scales. The new feature bank was again submitted to UMAP for dimensionality reduction, independently for all 10 participants. This procedure generated a two-dimensional feature space unique to each participant recapitulating the organization of the amygdala. We then plotted the values generated by UMAP onto their original coordinate points inside the amygdala in the native qT1 space of each subject for further visualization and analysis.

To finally compare subject-specific in vivo MRI findings to our previously defined histological components, we first co-registered each subject's qT1 scan to the ICBM152 template and applied the resulting transform to each participant's U1 and U2 map. We applied existing transformations to bring the histological U1 and U2 spaces to the same template space (*Xiao et al., 2019*; *Paquola et al., 2021*). This allowed us to contrast the UMAP components of all 10 subjects and BigBrain, along the same *x*, *y*, and *z* coordinate axes.

## rsfMRI image processing and analysis

Processing employed micapipe v.0.2.3 (*Cruces et al., 2022*), which combines functions from AFNI (*Cox, 1996*), FastSurfer (*Henschel et al., 2020*; *Henschel et al., 2022*; *Faber et al., 2022*), workbench command (*Marcus et al., 2011*), and FSL (*Jenkinson et al., 2012*). Images were reoriented, as well as motion and distortion corrected. Motion correction was performed by registering each timepoint volume to the mean volume across time points, while distortion correction utilized main phase and reverse phase encoded field maps. Leveraging our multi-echo acquisition protocol, nuisance variable signals were removed with tedana (*DuPre et al., 2021*). Volumetric timeseries were averaged for registration to native FastSurfer space using boundary-based registration (*Greve and Fischl, 2009*), and mapped to individual surfaces using trilinear interpolation. Cortical timeseries were mapped to the hemisphere-matched fs-LR template using workbench tools then spatially smoothed with a 10 mm Gaussian kernel. Surface- and template-mapped cortical timeseries were corrected for motion spikes using linear regression of motion outliers provided by FSL. Functional and anatomical spaces were co-registered using label-based affine registration (*Avants et al., 2010*) and the SyN algorithm available in ANTs (*Avants et al., 2011*).

## Functional network mapping of amygdala microstructural subregions

We averaged the rsfMRI timeseries within amygdala subregions defined by the highest and lowest 25% of values in each participant's own qT1-derived U1 component. Functional connectivity of each amygdala subregion to the rest of the cortex was determined using Pearson correlation between the timeseries of each amygdala subregion and each cortical vertex. Resulting correlation coefficients underwent Fisher R-to-Z transformation to increase the normality of the distribution of functional connectivity values.

A mixed effects model implemented with the BrainStat toolbox (*Larivière et al., 2023*) assessed differences between the cortical connectivity profiles of each U1 subregion, while considering age, and sex, and fixed effects and subject identity as a random effect. We corrected findings for family-wise errors (FWE) using random field theory ($p_{FWE}$ <0.05; cluster-defining threshold (CDT)=0.01). We further contextualized differences in each connectivity map by decoding connectivity profiles with functional activation maps aggregated in NeuroSynth (*Yarkoni et al., 2011*) and made available via BrainStat (*Larivière et al., 2023*). Our approach contrasted the overall connectivity profile of the amygdala to those of each of its microstructurally-defined subregions. First, we divided subregional amygdala-cortical connectivity profiles into seven established functional network communities (*Yeo et al., 2011*) and contrasted their average connectivity strength across each network. Meta-analytic functional decoding of the connectivity patterns of both amygdala subregions and the whole amygdala also highlighted different cognitive affiliations of each seed. Spatial correlations between the amygdala's cortical connectivity profile and spatial activation maps associated with each term allowed

us to retain ten terms associated with different cognitive domains. We then compared the association between the activation patterns of retained terms and the cortical connectivity profiles of each amygdala U1 subregion.

## Acknowledgements

HA acknowledges funding from the Fonds de la Recherche du Québec – Nature et Technologie (FRQNT) Master's Training Scholarship DGC acknowledges support from FRQ-Sante and Savoy Foundation JR acknowledges support from CIHR BCB acknowledges support from NSERC, CIHR, SickKids Foundation, BrainCanada, Future Leaders Research Grant, Helmholtz International BigBrain Analytics and Learning Laboratory (HIBALL), Healthy Brains and Healthy Lives, FRQS, and the Canada Research Chairs program.

## Additional information

### Funding

| Funder | Grant reference number | Author |
| --- | --- | --- |
| Fonds de recherche du Québec – Nature et technologies | | Hans Auer |
| Fonds de Recherche du Québec - Santé | Doctoral Fellowship | Donna Gift Cabalo |
| Savoy Foundation | Doctoral Scholarship | Donna Gift Cabalo |
| Canadian Institutes of Health Research | FDN-154298 | Boris C Bernhardt |
| Natural Sciences and Engineering Research Council of Canada | Discovery | Boris C Bernhardt |
| Sickkids Research Institute | | Boris C Bernhardt |
| Fondation Brain Canada | | Boris C Bernhardt |
| Helmholtz International BigBrain Analytics and Learning Laboratory | | Boris C Bernhardt |
| Healthy Brains for Healthy Lives | | Boris C Bernhardt |
| Canada Research Chairs | Cognitive Neuroinformatics of Heathy and Diseased Brains | Boris C Bernhardt |
| Canadian Institutes of Health Research | Fellowship | Jessica Royer |
| Canadian Institutes of Health Research | PJT-174995 | Boris C Bernhardt |
| Canadian Institutes of Health Research | PJT-191853 | Boris C Bernhardt |

The funders had no role in study design, data collection and interpretation, or the decision to submit the work for publication.

### Author contributions

Hans Auer, Data curation, Software, Formal analysis, Validation, Investigation, Visualization, Writing – original draft, Writing – review and editing; Donna Gift Cabalo, Raúl Rodríguez-Cruces, Resources, Data curation, Software, Writing – review and editing; Oualid Benkarim, Resources, Software, Methodology, Writing – review and editing; Casey Paquola, Conceptualization, Methodology, Writing

– original draft, Writing – review and editing; Jordan DeKraker, Data curation, Software, Visualization, Writing – review and editing; Yezhou Wang, Data curation; Sofie Louise Valk, Methodology, Writing – original draft, Writing – review and editing; Boris C Bernhardt, Conceptualization, Supervision, Funding acquisition, Investigation, Visualization, Methodology, Writing – original draft, Project administration, Writing – review and editing; Jessica Royer, Conceptualization, Software, Formal analysis, Supervision, Validation, Investigation, Visualization, Methodology, Writing – original draft, Project administration, Writing – review and editing

### Author ORCIDs
Casey Paquola ⓘD https://orcid.org/0000-0002-0190-4103
Jordan DeKraker ⓘD https://orcid.org/0000-0002-4093-0582
Sofie Louise Valk ⓘD https://orcid.org/0000-0003-2998-6849
Boris C Bernhardt ⓘD https://orcid.org/0000-0001-9256-6041
Jessica Royer ⓘD https://orcid.org/0000-0002-4448-8998

### Ethics
For BigBrain, the postmortem brain was acquired through the body donor program of the University of Düsseldorf in accordance with legal requirements (Amunts et al., 2013). For MICA-PNI, the MRI data acquisition protocols were approved by the Research Ethics Board of McGill University. All participants provided written informed consent, which included a provision for openly sharing all data in anonymized form (Cabalo et al., 2024).

Reviewer #1 (Public review): https://doi.org/10.7554/eLife.101950.3.sa1
Reviewer #2 (Public review): https://doi.org/10.7554/eLife.101950.3.sa2
Author response https://doi.org/10.7554/eLife.101950.3.sa3

## Additional files

### Supplementary files
Supplementary file 1. See a to d for tables cited in this study.

MDAR checklist

### Data availability
Analysis notebooks related to this project are available on GitHub (https://github.com/MICA-MNI/micaopen/tree/master/AmygdalaUMAP copy archived at *Auer and Royer, 2025*). BigBrain data are available on (https://osf.io/xkqb3/), 7T data are available on the Open Science Framework (https://osf.io/mhq3f/).

The following previously published datasets were used:

| Author(s) | Year | Dataset title | Dataset URL | Database and Identifier |
|---|---|---|---|---|
| Xiao Y, Lau JC, Anderson T, DeKraker J, Collins DL, Peters TM, Khan AR | 2023 | Accurate registration of the BigBrain dataset with the MNI PD25 and ICBM152 atlases | https://doi.org/10.17605/OSF.IO/XKQB3 | Open Science Framework, 10.17605/OSF.IO/XKQB3 |
| Cabalo DG, Rodriguez-Cruces R, Bernhardt BC, DeKraker J, Royer J, Leppert I, Thevakumaran R, Hwang Y, Kebets V, Tavakol S, Wang Y, Zhou Y, Benkarim O, Eichert N, Paquola C, Tardif CL, Rudko D, Smallwood J | 2024 | MICA-PNI: Precision NeuroImaging and Connectomics | https://doi.org/10.17605/OSF.IO/MHQ3F | Open Science Framework, 10.17605/OSF.IO/MHQ3F |

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
