## [Editor Report · eLife Assessment]

This **valuable** contribution combines high-resolution histology with magnetic resonance imaging in a novel way to study the organisation of the human amygdala. The main findings **convincingly** show the axes of microstructural organisation within the amygdala and how they map onto the functional organisation. Overall, the approach taken in this paper showcases the utility of combining multiple modalities at different spatial scales to help understand brain organisation.

---

## [Referee Report · Reviewer #1 (Public review)]

The paper by Auer et. makes several contributions:

(1) The study developed a novel approach to map the microstructural organization of the human amygdala by applying radiomics and dimensionality reduction techniques to high-resolution histological data from the BigBrain dataset.

(2) The method identified two main axes of microstructural variation in the amygdala, which could be translated to in vivo 7 Tesla MRI data in individual subjects.

(3) Functional connectivity analysis using resting-state fMRI suggests that microstructurally defined amygdala subregions had distinct patterns of functional connectivity to cortical networks, particularly the limbic, frontoparietal, and default mode networks.

(4) Meta-analytic decoding was used to suggest that the superior amygdala subregion's connectivity is associated with autobiographical memory, while the inferior subregion was linked to emotional face processing.

(5) Overall, the data-driven, multimodal approach provides an account of amygdala microstructure and possibly function that can be applied at the individual subject level, potentially advancing research on amygdala organization.

---

## [Referee Report · Reviewer #2 (Public review)]

Summary:

This study bridges a micro- to macroscale understanding of the organization of the amygdala. First, using a data-driven approach, the authors identify structural clusters in the human amygdala from high-resolution post-mortem histological data. Next, multimodal imaging data to identify structural subunits of the amygdala and the functional networks in which they are involved. This approach is exciting because it permits the identification of both structural amygdalar subunits, and their functional implications, in individual subjects. There are, however, some differences in the macro and microscale levels of organization that should be addressed.

Strengths:

The use of data-driven parcellation on a structure that is important for human emotion and cognition, and the combination of this with high-resolution individual imaging-based parcellation, is a powerful and exciting approach, addressing both the need for a template-level understanding of organization as well as a parcellation that is valid for individuals. The functional decoding of rsfMRI permits valuable insight into the functional role of structural subunits. Overall, the combination of micro to macro, structure, and function, and general organization to individual relevance is an impressive holistic approach to brain mapping.

---

## [Author Response]

The following is the authors’ response to the original reviews.

**Reviewer #1:**
The paper by Auer et. makes several contributions: (1) The study developed a novel approach to map the microstructural organization of the human amygdala by applying radiomics and dimensionality reduction techniques to high-resolution histological data from the BigBrain dataset. (2) The method identified two main axes of microstructural variation in the amygdala, which could be translated to in vivo 7 Tesla MRI data in individual subjects. (3) Functional connectivity analysis using resting-state fMRI suggests that microstructurally defined amygdala subregions had distinct patterns of functional connectivity to cortical networks, particularly the limbic, frontoparietal, and default mode networks. (4) Meta-analytic decoding was used to suggest that the superior amygdala subregion's connectivity is associated with autobiographical memory, while the inferior subregion was linked to emotional face processing. (5) Overall, the data-driven, multimodal approach provides an account of amygdala microstructure and possibly function that can be applied at the individual subject level, potentially advancing research on amygdala organization.

We thank the Reviewer for the positive comments and insightful evaluation of the work.

(1.1) Although these are meritorious contributions there are some concerns that I will summarize below. The paper makes little-to-no contact with the monkey literature regarding the anatomy of amygdala subregions, their functionality, and their patterns of anatomical connectivity. This is surprising because such literature on non-human primates is a very important starting point for understanding the human amygdala. I recommend taking a careful look at the work by Helen Barbas, among others. There are too many papers to cite but a notable example is: Ghashghaei, H. T., Hilgetag, C. C., & Barbas, H. (2007). Sequence of information processing for emotions based on the anatomic dialogue between prefrontal cortex and amygdala. Neuroimage, 34(3), 905-923. The work of Amaral is also highly relevant.

As suggested, we included the important work of Amaral et al. as well as Ghashghaei et al. highlighting its contribution to mapping the intricate anatomy and function of the amygdala in non-human primates. We comment on this in the Introduction of the manuscript. Please see P.3.

“Early research on the amygdala in non-human primates has been instrumental in understanding its intricate structure, function and patterns of anatomical connectivity (Amaral and Price 1984; Ghashghaei et al. 2007). This foundational study highlights the amygdala’s different subdivisions, most notably the basomedial nucleus (BM), basolateral nucleus (BL), and central nucleus (Ce) (Amaral et al. 1992). Furthermore, this work describes a dense network between these subdivisions and the prefrontal cortex, most strongly found in the posterior orbitofrontal and anterior cingulate areas.”

(1.2) Furthermore, the authors subscribe to a model with LB, CM, and SF sectors. How does the SF sector relate to monkey anatomy?

The overall organization of these subregions is largely conserved between humans and monkeys, reflecting their evolutionary relationship. While the basic subregional organization is conserved, there are still some important structural and functional differences between human and monkey amygdalae. For example, the SF subregion, often described in humans includes parts of the cortical nuclei (VCo), anterior amygdaloid area (AAA), amygdalohippocampal transition area (AHi), amygdalopiriform transition area (APir) as well as the lateral olfactory tract (LOT). This remark was added in the Discussion, on P.12:

“However, this region has been previously described as consisting of three main subdivisions: LB, CM, and SF, each composed of smaller subnuclei with distinct connectivity patterns and functions (Amunts et al. 2005; Ball et al. 2007; Bzdok et al. 2013; de Olmos and Heimer 1999). These subregions are largely conserved between humans and monkeys, reflecting their evolutionary relationship. However, there are still some considerable differences such as in the SF subregion, where its description in monkeys additionally contains the lateral olfactory tract (LOT) (De Olmos 1990).”

(1.3) The authors use meta-analytical decoding via NeuroSynth. If the authors like those results of course they should keep them but the quality of coordinate reporting in the literature is insufficient to conclude much in the context of amygdala subregion function in my opinion. I believe the results reported are at most "somewhat suggestive".

We agree with the Reviewer that use of data from NeuroSynth poses unique challenges, particularly relating to investigations of a small structure such as the amygdala. However, to clarify, these analyses decode the cortex-wide functional connectivity patterns of amygdala subregions and not activations within subregions defined by our microanatomical analyses. Additionally, comments from Reviewer 2 suggested expanding the NeuroSynth decoding to the contralateral hemisphere. As such, we decided to keep this analysis in the main manuscript but rephrase the interpretation of these findings in the Discussion to emphasize their exploratory nature on P.13:

“Functional decoding of subregional functional connectivity patterns indicated possible dissociations in cognitive (e.g., memory) and affective (e.g., emotional face processing) functions of the amygdala, echoing previous accounts of this region’s involvement in associative processing of emotional stimuli. Notably, these findings link the functional connectivity profile of a subregion partially co-localizing with LB to emotional face processing. The LB subregion has been previously linked to associative processing related to the integration of sensory information (Bzdok et al. 2013; Ghods-Sharifi, St Onge, and Floresco 2009; Pessoa 2010; Winstanley et al. 2004; Boyer 2008), which is consistent with the association with visual emotional information processing identified in the present work.”

(1.4) Another significant concern has to do with the results in Figure 3. The red and yellow clusters identified are quite distinct but the differences in functional connectivity are very modest. Figure 3C reveals very similar functional connectivity with the networks investigated. This is very surprising, and the authors should include a careful comparison with related findings in the literature. Overall, there is limited comparison between the observed results and those obtained via other methods. On a more pessimistic note, the results of Figure 3 seem to question the validity of the general approach.

We agree with the Reviewer that we can indeed observe considerable overlap between functional connectivity profiles of amygdala subregions. The amygdala is a relatively small structure, leading to likely interconnectivity between its subregions (Bzdok et al. 2013) in addition to considering BOLD signal autocorrelation within this region. In addition, functional signals in the amygdala are affected by relatively lower signal-to-noise ratio (SNR), a limitation extending to temporobasal and mesiotemporal regions. Despite these challenges, our technique remained sensitive to detect subtle differences in connectivity patterns even in this small group of subjects in this restricted subcortical territory.

In the revised manuscript, we further highlight these caveats in the Discussion (P.13):

“Although these findings are promising, we also observe considerable overlap between functional connectivity networks of both our defined subregions. Indeed, the amygdala is a relatively small structure, leading to likely interconnectivity between its subregions and locally high signal autocorrelation. Functional connectivity and microstructure in the amygdala are certainly related, however previous work suggests they do not perfectly overlap (Bzdok et al. 2013). In addition, this region is affected by relatively low signal-to-noise ratio (SNR), as is observed in broader temporobasal and mesiotemporal territories.”

(1.5) Some statements in the Discussion feel unwarranted. For example, "significant dissociation in functional connectivity to prefrontal structures that support self-referential, reward-related, and socio-affective processes." This feels way beyond what can be stated based on the analyses performed.

We agree that this interpretation may reach beyond the analyses performed and reported findings. We have adjusted this portion of the text accordingly in our Discussion on functional connectivity findings (P.13):

“Qualitatively, we found that the subregion defined by the highest 25% of U1 values mainly overlapped with what is commonly defined as the superficial and centromedial subregions, whereas the lowest 25% U1 values subregion overlapped mostly with the laterobasal division. Interestingly, CM and SF characterized subregions showed significantly stronger functional connectivity to prefrontal structures. This finding aligns with previous work demonstrating unique affiliations between the CM subregion and anterior cingulate and frontal cortices (Kapp, Supple, and Whalen 1994; Barbour et al. 2010), as well as between the SF subregion and the orbitofrontal cortex (Goossens et al. 2009; Caparelli et al. 2017; Pessoa 2010; Klein-Flügge et al. 2022).”

Additionally, we have also edited our Discussion to ensure that our interpretations are grounded in the analyses conducted, while framing the findings as potential avenues for future work. Please see P.13.

“Functional decoding of functional connectivity results indicated possible dissociations in cognitive (e.g., memory) and affective (e.g., emotional face processing) functions of the amygdala, echoing previous accounts of this region’s functional specialization and subregional segregation of associative processing of emotional stimuli.”

**Recommendations for the authors:**
(1.6) Figure 1 has panels A-I but only A-D are discussed in the caption. The orientation of the slices is not indicated which makes it very hard to follow for most readers.

The subpanels are now referred to in the revised Results. We also added a notation on the orientation of the slices and described them accordingly in our Figure 1 description. (P.5-6):

“(A) The amygdala was segmented from the 100-micron resolution BigBrain dataset using an existing subcortical parcellation (Xiao et al. 2019). Slice orientation is consistent across all panels in this figure.”

(1.7) Some figure references in the text seem to be incorrect; please check that the text refers to the correct figure number and panel.

We thank the Reviewer for pointing this out. We thoroughly revised the correspondence between figure panel labels and their referencing in the text.

**Reviewer #2:**
This study bridges a micro- to macroscale understanding of the organization of the amygdala. First, using a data-driven approach, the authors identify structural clusters in the human amygdala from high-resolution post-mortem histological data. Next, multimodal imaging data to identify structural subunits of the amygdala and the functional networks in which they are involved. This approach is exciting because it permits the identification of both structural amygdalar subunits, and their functional implications, in individual subjects. There are, however, some differences in the macro and microscale levels of organization that should be addressed.Strengths:The use of data-driven parcellation on a structure that is important for human emotion and cognition, and the combination of this with high-resolution individual imaging-based parcellation, is a powerful and exciting approach, addressing both the need for a template-level understanding of organization as well as a parcellation that is valid for individuals. The functional decoding of rsfMRI permits valuable insight into the functional role of structural subunits. Overall, the combination of micro to macro, structure, and function, and general organization to individual relevance is an impressive holistic approach to brain mapping.

We thank the Reviewer for their constructive and helpful feedback on our work.

Weaknesses:(2.1) UMAP 1, as calculated from the histological data, appears to correlate well across individuals, and decently with the MRI data, although the medial-lateral coordinate axis is an outlier. UMAP 2, on the other hand, does not appear to correlate well with imaging data or across individuals. This does pose a problem with the claim that this paper bridges micro- and macroscale parcellations. One might certainly expect, however, that different levels of organization might parcellate differently, but the authors should address this in the discussion and offer ways forward.

Data driven methods hold several advantages for the quantitative extraction of signal from the underlying data in an observer-independent manner. However, these techniques are also sensitive to potential idiosyncrasies in the data. In the present work, our main analyses rely on the processing of a histological dataset (BigBrain) providing a unique opportunity for high-resolution analysis of amygdala histology and in vivo translation of findings leveraging ultra-high field MRI (n=10). However, both datasets are limited by their small sample size (n=1 for BigBrain and n=10 for MICA-PNI). As a result, we speculate that signal variations captured by U2 may be sensitive to artifacts or subject-specific sources of variance. Moving forward, this hypothesis could be assessed in future work via the analysis of larger histological and neuroimaging datasets to better track recurring features picked up by U2 or the association of these unique topographies with behavioural markers.

As suggested, we included a section in our Discussion highlighting this shortcoming and the importance for larger datasets moving forward. Please see P.11-12.

“However, it is important to note that both datasets analyzed in this work are limited by their small sample size (n=1 for BigBrain and n=10 for MICA-PNI). We speculate that the signal variations captured by U2 may be sensitive to artifacts or subject-specific sources of variance, potentially explaining why it was not consistent between subjects and modalities. Moving forward, this hypothesis could be assessed in future work via the analysis of larger histological and neuroimaging datasets to better track recurring features picked up by U2 or the association of these unique topographies with behavioural markers.”

(2.1) It would be interesting to see functional decoding for the right amygdala. This could be included in the supplementary material. A discussion of differences in the results in the two hemispheres could be illuminating.

In accordance with the Reviewer’s suggestion, we added Supplementary figure S2 exploring the decoding of connectivity profiles of the right amygdala stratified by its cytoarchitectural embedding with UMAP.

Upon analysis, dissociation in functional connectivity patterns over the right amygdala were less evident, leading to overall similar functional decoding across the two clusters. We refer to this Supplementary Figure in our Discussion on P.13.

“For the right amygdala, dissociation in functional connectivity patterns were more subtle, leading to overall similar functional decoding across the two clusters. (Figure S2)”

(2.3) The authors acknowledge that this mapping matches some but not all subunits that have been previously described in the amygdala. It would be helpful to neuroanatomists if the authors could discuss these differences in more detail in the discussion, to identify how this mapping differs and what the implications of this are.

In our work, we focus on mapping the three well characterized amygdala subregions, specifically the superficial (SF), centromedial (CM) and laterobasal (LB) subdivisions. Qualitative histological accounts have indeed delineated multiple subunits within these subregions which we now describe in the revised manuscript. Due to the lower resolution of in vivo MRI data used in this work relative to post mortem histology, we focused our analyses on larger subregions that could be more reliably mapped to native quantitative T1 spaces of each participant. We now overview this issue in the Discussion. Please see P.12.

“Although qualitative histological accounts have indeed delineated multiple subunits within these general regions, the present work focuses on three subdivisions (Amunts et al. 2005) to account for resolution disparities when translating our findings to in vivo MRI data. The LB subdivision includes the basomedial nucleus (Bm), basolateral nucleus (BL), lateral nucleus (LA) and paralaminar nucleus (PL). Moving medially, the CM subdivision includes the central (Ce) and medial nuclei (Me), while the SF subdivision includes the anterior amygdaloid area (AAA), amygdalohippocampal transition area (AHi), amygdalopiriform transition area (APir), and ventral cortical nucleus (VCo) (Heimer et al. 1999). However, disagreement on the precise attribution of nuclei to broader subdivisions motivated our investigations of probabilistic subunits of the amygdala (Kedo et al. 2018). The development of new tools to segment amygdala subnuclei in vivo offers opens opportunities for future work to further validate our framework at the precision of these nuclei within subjects (Saygin et al. 2017).”

(2.4) The acronym UMAP is not explained. A brief explanation and description would be useful to the reader.

We moved the expanded acronym from the *Methods* to the first instance of the term UMAP in our paper, found in the *Introduction*. As suggested, we also added a sentence describing the technique. Please see P.6.

“We then applied Uniform Manifold Approximation and Projection (UMAP), a non-linear dimensionality reduction technique that preserves the local and global structure of high-dimensional data by projecting it into a lower-dimensional space (Becht et al. 2018), to the resulting 20-feature matrix to derive a 2-dimensional embedding of amygdala cytoarchitecture (Figure 1D).”